# Implementing an intervention to enhance care delivery and consistency for people with hip fracture and cognitive impairment in acute hospital wards: a mixed methods process evaluation of a randomised controlled feasibility trial (PERFECTED)

Tamara Backhouse ![ORCID],[1] Chris Fox,[2,3] Simon P Hammond ![ORCID],[4] Fiona Poland,[1] Vicki McDermott-Thompson,[5] Bridget Penhale,[1] Jane L Cross ![ORCID] [1]

For numbered affiliations see end of article.

**Correspondence to**
Dr Jane L Cross;
j.cross@uea.ac.uk

## ABSTRACT

**Objectives** To determine how, and under what circumstances, the PERFECT-ER intervention was implemented in five acute hospital wards and impacted on staff practices and perceptions.

**Design** Mixed methods process evaluation (undertaken between 2016 and 2018).

**Setting** Five acute hospital wards across three different UK regions.

**Participants** Patients (n=3) admitted to acute wards with hip fracture and cognitive impairment, their relatives (n=29) and hospital staff (n=63).

**Interventions** PERFECT-ER, a multicomponent intervention designed to enhance the recovery of patients with hip fracture and cognitive impairment was implemented for 18 months. PERFECT-ER was implemented at ward level ensuring that multiple new and existing practices were undertaken consistently, on the assumption that collectively, small individual advances would improve care delivery for patients.

**Primary and secondary outcome measures** Implementation of the PERFECT-ER intervention examined through regular intervention scores, service improvement staff reports and action plans, and semi-structured interviews and focus groups.

**Results** The process evaluation identified points of implementation vulnerability and strength. All wards implemented some elements of PERFECT-ER. Implementation was fragile when ward pressures were high and when ward staff perceived the relative priority of intervention practices to be low. Adaptations to the implementation process may have reduced whole-ward staff engagement with implementation. However, strategical enlistment of senior ward influencers (such as ward managers, orthogeriatricians) combined with service improvement lead in-ward peer pressure tactics facilitated implementation processes.

**Conclusions** Our study suggests that implementation was expedited when senior staff were on board as opinion

## STRENGTHS AND LIMITATIONS OF THIS STUDY

⇒ Strengths of this study include obtaining data from multiple sources and stakeholders from five different acute hospital wards across seven implementation cycles.
⇒ Another strength was the independence of the core process evaluation team from the feasibility trial team.
⇒ Due to the nature of the intervention, documentary evidence from patients' notes was used by improvement leads to complete intervention checklists and this information could not be verified by researchers.
⇒ The research team could not confirm whether implementation was due to changes in clinical care, documenting practices or due to improvement leads' own actions.

leaders and formally appointed internal implementation leaders exerted their power. Within hierarchical settings such as acute wards, key individuals appeared to influence implementation through endorsement and sometimes enforcement. This indicates that whole-ward interventions may not always require cognitive engagement from all ward staff to implement changes. Future ward-level implementation studies could consider how best to engage staff and most importantly, which staff to best target.

**Trial registration number** ISRCTN99336264.

## BACKGROUND

The increasing incidence of dementia is a worldwide public health challenge, requiring multiple strategies including action to improve the quality of healthcare services.[1] Providing acute care for patients with cognitive impairment (including those with either diagnosed or assumed dementia and/or

delirium) is challenging for hospital staff.[2 3] Patients with cognitive impairment have complex needs related to delirium, behaviours that challenge, incontinence, dehydration and communication difficulties. These make necessary medical, nursing and rehabilitation processes more difficult for staff to undertake.[2 4–7]

In acute settings, over 40% of the people with a hip fracture have cognitive impairment.[8] Usual biomedical approaches[9] and limited evidence for rehabilitation for these patients[10–12] create further difficulties for staff. Consequently, patients with hip fracture and cognitive impairment have increased risk of unfavourable outcomes,[13–16] impaired rehabilitation and recovery,[6 17] with a worse prognosis than patients without cognitive impairment.[8 18] Hospital staff need to manage these complexities and complex multifaceted interventions to improve quality in healthcare delivery, and outcomes are increasingly required.[19]

The PERFECTED research programme was funded by the National Institute for Health Research (NIHR), UK, to develop and test an intervention to enhance the recovery of patients with hip fracture and cognitive impairment on acute wards. PERFECT-ER is a complex multicomponent service improvement intervention which aims to enhance recovery using multiple marginal gains in the quality of care.[20–23] Marginal gains include the implementation of new practices, standardisation of existing practices, ensuring consistency and cultivating a whole-ward 'in-it-together' ethos.

The PERFECT-ER intervention comprises a structured checklist,[24] Service Improvement Lead (SIL), Process Lead (PL) and implementation manual using a Plan, Do, Study, Act (PDSA) approach to facilitate behaviour change[25 26] (see logic model, online supplemental figure 1). PERFECT-ER is a whole ward approach involving all staff delivering patient care. The PERFECT-ER checklist

has 68 practice items (including clinical, functional, psychosocial and cognitive elements), which span admission, preoperative, postoperative, rehabilitation and discharge periods and 15 organisational items, which highlight the policies behind current practices in the hospital. Checklists have been used before to improve the quality of care,[27 28] however such lists rarely change practice unless combined with mechanisms for change within the socio-practice contexts of hospital settings.[29 30] For example, much relational work in hospitals relies on individual staff members and supportive organisational structures.[31]

PERFECT-ER is implemented by an SIL, a formally appointed internal implementation lead, a qualified healthcare professional, based day-to-day on the selected ward. They attend a 3-day induction training in intervention delivery. They were supported by a senior doctor acting as a PL to facilitate implementation across the organisation. The checklist, designed to overlay current hip fracture pathways, identifies areas of strength and weaknesses in current practice. It is used to score a sample of patients' notes (documentary evidence) to determine current practice and identify areas for improvement. Scores are shared with staff at action-planning meetings, where discussions create shared action plans to address lower scoring items. The Plan (action plan), Do (implementation), Study (fill in checklist and assess scores), Act (engage ward staff) cycle then recommences.

As part of the wider PERFECTED programme of research, we conducted a process evaluation of the PERFECT-ER feasibility trial.[20 23] The aim of this process evaluation was to determine how and under what circumstances the PERFECT-ER intervention was implemented in acute hospital wards and impacted on staff practices and perceptions (for published protocol, see online supplemental material).[20 32–34]

| | PERFECT-ER checklist scores *n* | SIL action plans *n* | SIL reflective reports *n* | Patient interviews *n* | Relative interviews *n* | Ward staff participants (interviews/ focus groups) *n* |
|---|---|---|---|---|---|---|
| Ward A | 8 | 7 | 21 | 2 | 8 | 10 |
| Ward B | 8 | 7 | 21 | 0 | 5 | 13 |
| Ward C | 8 | 6 | 20 | 1 | 5 | 12 |
| Ward D | 8 | 6 | 15 | 0 | 4 | 15 |
| Ward E | 8 | 5 | 15 | 0 | 7 | 13 |
| Total | 40 | 31 | 92 | 3 | 29 | 63 |

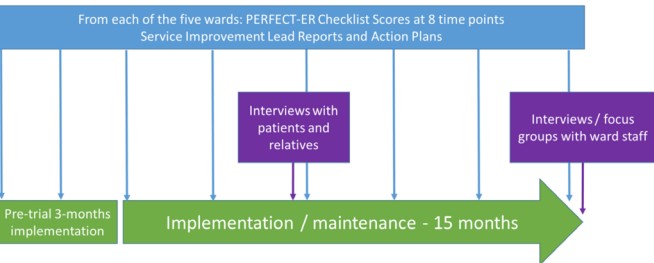

**Figure 1** Process evaluation data collection. SIL, Service Improvement Lead.

## METHODS

Qualitative data from process evaluations of trials are widely used to generate understandings of intervention implementation[35] and can be of critical importance.[36] However, there is limited qualitative evidence about the implementation of complex interventions to enhance recovery in hospital settings.[37] Therefore, to examine implementation in acute wards in detail, we used a mixed methods approach incorporating qualitative data.

The process evaluation took place in five different acute National Health Service hospital wards in five separate geographical areas across England and Scotland where the PERFECT-ER intervention was being implemented. Resource use and patient and relative outcomes are reported elsewhere.[23] We used the Medical Research Council (MRC) guidance[32 33 38–41] for the evaluation and the Standards for Reporting Implementation Studies.[42]

Process evaluation data were gathered between August 2016 and March 2018 over seven PDSA cycles (see figure 1 for data collected from each ward). We collected

PERFECT-ER checklist scores, SIL-generated reflective reports of implementation activities and action plans and semi-structured interview and focus group data from each ward. PERFECT-ER checklist scores were obtained from patient notes by the SILs. At each ward there were eight checklist time points: 3 months pre-trial, 1.5 months pre-trial, trial baseline, 4 months, 7 months, 10 months, 13 months and end of trial (15 months).

We undertook: semi-structured face-to-face or telephone interviews with hospital staff and relatives of patients; focus groups with hospital staff; and face-to-face interviews with patients. We purposively sampled ward-based staff (up to 15 from each ward) who had been in contact with the intervention. Interviews and focus groups with staff investigated their knowledge of, and engagement with, the intervention and implementation processes. These were undertaken after trial processes finished to prevent interviews impacting on implementation. Patients and relatives were offered the opportunity to take part in interviews by a research nurse during a follow-up visit for the trial. Interviews were designed to explore patients and relatives' experiences of the intervention ward. All had capacity to consent. Patients and relatives' interviews took place in their home or in a private room at the hospital according to their wishes. Interviews and focus groups were conducted by academic researchers not collecting data or consent for the feasibility trial (SPH, TB and VM-T).

### Patient and public involvement

PERFECTED prioritised public and patient involvement (PPI) from grant development to completion. PPI representatives provided feedback on participant information materials and the study protocol, and had roles on the Trial Steering Committee, Data Monitoring Ethics Committee and Advisory Groups. For the process evaluation, PPI carer representatives co-interviewed, with researchers, in six interviews with relatives. They received training and support to facilitate this.

### Analysis

All interviews and focus groups were digitally recorded and transcribed verbatim. Identifying features were removed to ensure confidentiality and anonymity. Qualitative analysis was iterative throughout the data collection and thematic, focusing on implementation processes informed by the MRC Process Evaluations of Complex Interventions[37–39] including dose, reach, fidelity, adaptations, mechanisms of impact and context and the Consolidated Framework for Implementation Research (CFIR)[43] in regard to aspects of the inner setting (hospital ward), characteristics of individuals and process. NVivo software was used to manage the data. TB generated initial codes from repeated reading of data, identifying tentative themes, before multiple analysis meetings with the study team (JLC, SPH, FP, VM-T and CF), which refined and developed these analyses both across sites and within individual sites. Quantitative data from PERFECT-ER checklist scores, collected from five patients at each of the five sites at eight time points, were analysed descriptively by ward and implementation cycle.

### RESULTS

Sixty-nine interviews and 10 focus groups were conducted, table 1 describes this population.

### Dose

Figure 2 shows PERFECT-ER checklist scores for each ward by percentage of elements delivered throughout the study. All wards implemented some elements of PERFECT-ER. Dose varied across wards. Availability of resources appeared to help or hinder implementation. Wards B and C implemented well (PERFECT-ER scores increasing 22% and 12%, respectively, over the study). Wards B and C had dedicated SIL time and clear ward processes which could be used for implementation. Wards A (PERFECT-ER scores increasing 8%), D (PERFECT-ER scores decreasing 7%) and E (PERFECT-ER scores

**Table 1** Participant demographics

| Participant | Total n=95 n (%) | Female n (%) | Age mean (range) | Method of data collection by n | | |
| --- | --- | --- | --- | --- | --- | --- |
| | | | | Face-to-face interview | Telephone interview | Focus group |
| Patient | 3 (3) | 3 (100) | 78 (68–87) | 3 | 0 | 0 |
| Relatives* | 29 (31) | 22 (76) | 57 (22–75) | 7 | 22 | 0 |
| Service improvement lead | 6 (6) | 5 (83) | 45 (29–59) | 1 | 5 | 0 |
| Process lead (consultant geriatricians) | 5 (5) | 1 (20) | 44 (37–49) | 0 | 5 | 0 |
| Research nurse | 8 (8) | 8 (100) | 43 (26–54) | 0 | 6 | 2 |
| Hospital staff† | 44 (46) | 43 (98) | 39 (22–60) | 19 | 1 | 24 |

*Daughter (n=15), son (n=3), daughter-in-law (n=4), son-in-law (n=3), niece (n=1), granddaughter (n=1), husband (n=1) and sister-in-law (n=1).
†Nurse (n=12), senior nurse (n=9), healthcare assistant (n=8), occupational health therapist (n=4), physiotherapist (n=3), ward administrator (n=2), ward housekeeper (n=2), doctor (n=1), trauma assistant (n=1), general therapy assistant (n=1) and clinical trial assistant (n=1).

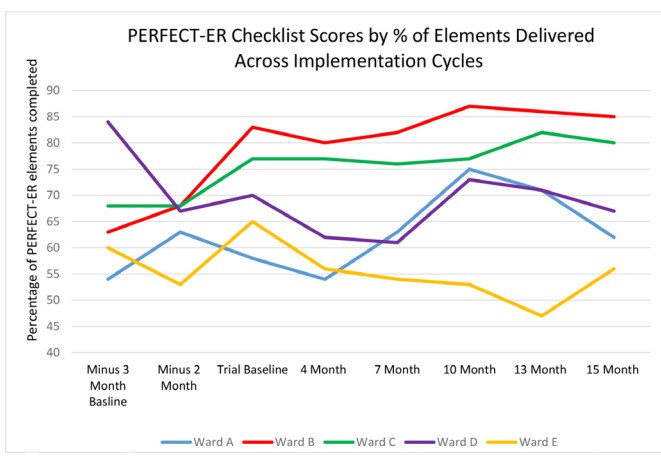

**Figure 2** PERFECT-ER checklist scores by percentage and implementation site.

decreasing 9%) struggled with severe ward pressures, including staff shortages, affecting implementation. Wards D and E also had difficulties providing dedicated SIL time and the implementation climate reflected a lack of consistent support from ward staff. Ward D had a different SIL to obtain the baseline data from the rest of the cycles. Average dose delivered was 68%. PERFECT-ER practice items least implemented were: (1) carers given the opportunity to accompany the patient in the recovery room, and (2) pain assessed daily using a pain scale specific to patients with cognitive impairment.

### Reach

A proportion of ward staff at all sites engaged with PERFECT-ER to some degree. Many staff at Ward C were largely unaware of the study and when asked about changes due to the intervention their typical response was:

> Nurse Practitioner: I wouldn't have thought, not through PERFECT-ER no, I wouldn't think so no.
>
> Senior Nurse: Or if it is through PERFECT-ER no-one has told us it has come from that. (Focus Group, Ward C)

Nevertheless, PERFECT-ER elements implemented for the study were now part of their usual daily practice and valued.

> Nurse Practitioner: …all the tools that we are bringing in are certainly useful to help appropriately nurse the patients that we actually do have (Focus Group, Ward C)

This suggests that implementation of PERFECT-ER elements did not rely on the engagement of all ward staff with the process.

In wards C and E several staff members said they would have liked more information about the intervention and an active role with implementation.

> Physiotherapist: Just to know more about it [PERFECT-ER] really… So, we could get more involved basically. (Focus Group, Ward E)

This suggests optimal reach was not always achieved. Nevertheless, across wards elements of PERFECT-ER were viewed as having improved staff awareness, particularly of pain and delirium.

> Occupational Therapist: I suppose it [PERFECT-ER] has highlighted it [pain] individually to me to be more aware and do something about it. (Focus Group, Ward C)

Interviews with relatives and patients suggested they did not notice an enhanced care experience. They expressed mixed experiences of care generally, but no-one noted PERFECT-ER specifically in their interview.

> By and large as I say it was a good experience umm I'd, I had done it before… I think it was better than it was 4 years ago. (Relative, Ward C)

> We felt like we were being a negative complaining family in a way because things just weren't going right. (Relative, Ward A)

Indicators of implementation may have reflected changes in documentary practices, and not changes in ethos or clinical practice as staff noted modifications to standardise recording and assessment practices.

> I don't think it changed my practice… it helped standardise those that perhaps wouldn't routinely do the same thing, but for those that were doing it, it allowed us to be clearly documented (PL, Ward E)

### Fidelity

Changes in ward ethos varied and appeared to be linked to the implementation climate, dose delivered and reach. Staff at ward C, where implementation went well, talked about changes in approach.

> Occupational Therapist: I think as [senior nurse] highlighted, it's [PERFECT-ER] brought an awareness of delirium… I have had a sense of that amongst the nursing colleagues …people are a lot more sensitive and patient and maybe provide better care maybe.
>
> Senior Nurse: Yeah I agree. (Focus Group, Ward C)

Some participants indicated PERFECT-ER provided a guiding philosophy for staff.

> PERFECT-ER is our ward philosophy do you know what I mean? … it actually is part of their [Ward staff's] umm vision part of their thinking about their patient care (SIL, Ward A)

Whereas, in Ward E which struggled to implement, there was no ethos change.

Occupational Therapist: As far as the PERFECTED Ward goes I am not sure how much it has really, it has sort of changed our practice as such because we kind of work to enable people anyway…

Lead Occupational Therapist: Yeah, I am still not a 100% clear about the different check lists …that makes it the PERFECTED Ward …I don't think it is something that is obvious what we are working towards. (Focus Group, Ward E)

Many aspects of PERFECT-ER were implemented as intended. However, the intervention was developed with one SIL intended as the appointed implementation leader. Whereas two wards employed three individuals to conduct the SIL role between them.

I do not feel I have been very supported by the other SILs …due to workload on the project ward and lack of staff. (SIL, Ward E)

…my management team, they tried to give an hour or two on occasional days …subject to how busy the ward was so found that majority of the times this ended up not being used. (SIL Report, Ward D)

In these wards, readiness for implementation was limited by organisational commitment of resources. SIL time was not protected and teamwork between SILs was lacking.

Two SILs from these wards (D and E) changed their own practices by ensuring PERFECT-ER documents were present in patients' notes instead of encouraging ward practices to change in a sustainable way.

…folders of these patients checked, and appropriate documentation placed in bedside folders. (SIL Report, Ward D)

The PERFECT-ER checklist was designed to be completed from documentary evidence, however two SILs, again from wards D and E, described using other evidence to complete their checklist.

…also standing at the end of the bed and looking … so I use quite a lot of things to get the information …you can find it in lots of different places and from obviously talking to staff as well. (SIL, Ward E)

### Adaptation

The PERFECT-ER implementation process was adapted significantly from that tested.[44] A ward-level action planning meeting was integral to the PDSA cycle, however none of the SILs held these. A few attempted some meetings.

…in the first sort of four or five months I had meetings with whatever staff I could grab …as time went on umm that got more and more difficult …staff movement …a lot of agency …pool staff (SIL, Ward B)

Ward pressures impacted on staff and their ability to get together as a team as part of implementation. Rather, SILs adapted the process using senior individuals (such as ward managers or PLs) or people relevant to checklist elements (eg, physiotherapists for rehabilitation items) to facilitate action planning.

I did manage to get some help from our orthogeriatrician [PL] …between the two of us we did a bit of a brainstorm on what we can implement (SIL, Ward D)

Thus, SILs undertook action planning with ward opinion leaders, but the missing team approach to action planning probably impacted on ward staff opportunities for engagement.

### Mechanisms of impact

Multiple mechanisms of impact were identified. A major strategy was peer pressure within the ward environment from SILs who enacted spot checks and audits of whether new or existing intervention practices were carried out, often followed up with reminders, 'nagging' and/or education.

Staff are aware that these need updating regularly and these are now part of the weekly ward audits to establish compliance. (SIL Report, Ward A)

…she just nagged us all the time. (Ward Manager, Ward A)

Spot check on new patient documentation to ensure that items are being used… Remains very hit and miss …ward staff identified and reminder letter given out. (SIL Report, Ward B)

Spent most of the day providing short education sessions to individual staff. (SIL Report, Ward C)

These actions suggest that SILs, as formally appointed internal implementation leaders, believed in their own self-efficacy in driving and accomplishing implementation goals.

Networks and communication were key to implementation work. SILs created and used networks; cascading and acquiring information via email correspondence, meetings, presentations to staff members, noticeboard use, education sessions and ongoing informal communication.

Good networking with other staff who are committed to providing a quality service. (SIL Report, Ward C)

Preparing for my presentation for the first of the ward time out days. I have a 45-min slot to discuss and promote PERFECT-ER (SIL Report, Ward A)

Ward managers played a key role as gatekeepers of change and SILs waited for their authorisation before implementing any PERFECT-ER elements.

Agreed to meet the senior charge nurse at next visit to get an agreement …and start the wheels in motion. (SIL Report, Ward C)

PERFECTED Process Evaluation: Lessons Learned

**Delivering consistency in intervention**

- Protected Service Improvement Lead time must be ensured, consider staff resource when recruiting sites
- If there is more than one Service Improvement Lead at a hospital site, each must have their own responsibilities/accountability
- Service Improvement Leads should be hired from the study ward or more time and support given to those external to the ward
- Consider administrative support for Service Improvement Leads (ward clerk time)
- Create a generic PERFECT-ER leaflet to provide a basic level of information about the intervention for ward staff and provide Service Improvement Lead's name and contact
- Provide Process Leads with role definition, clear responsibilities and make time allocation transparent
- Refine PERFECT-ER Manual to reflect feedback and role of opinion leaders in implementation

**Participant recruitment and retention**

- Research team to use a problem solving approach with Process Leads and research nurses to support recruitment and retention of participants
- Provide dementia training for research nurses
- Colour-code participant documents to reduce confusion
- Enable relative participants to fill in Case Report Forms/workbooks in own time rather than at the hospital – possible online option
- Reduce number of outcome measures to decrease burden on relatives and patients
- Consider care-home staff as informants if patients are discharged to a care home and relatives can no longer answer questions about the person's daily life

**Figure 3** PERFECTED: lessons learnt for a future definitive randomised controlled trial.

They and other key clinicians, such as PLs, facilitated implementation using their powerful positions as opinion leaders on the ward to endorse and enforce implementation.

> The ward sister [manager] has introduced this to the staff and put it in the 'Hot Topics' folder. (SIL Report, Ward B)

Two wards (A and E) enlisted PERFECT-ER 'Champions' to take on some responsibility and facilitate implementation while undertaking their usual roles.

> I then decided to have PERFECT-ER Champions… I can only do so much really …it is like 'right I want you to, you know, it is your responsibility to try and speak to your peers about this' (SIL, Ward A)

Implementation was enabled by aligning new elements of practice with current practices and organisational initiatives.

> …putting one of those charts in the Admission Pack …if it is not visible people won't think to use it, so we are trying to put it so as it is normal for them and they will grab a pack and everything is there ready (SIL, Ward E)

## Contextual factors

The study took place within the context of multiple pressures in all wards. Staffing issues were common. Ward A had extra beds on the ward, low staffing levels, and a lack of senior staff due to the ward manager's long-term sickness:

> This leaves the ward unsafe and staff are struggling to carry out their required duties never mind embrace change and add to their workload (SIL, Ward A)

This created barriers to implementing PERFECT-ER. Staff changes meant SILs needed to constantly engage with and inform new colleagues about PERFECT-ER practices on the ward.

The implementation climate in the inner setting involved a lack of interest from some staff members and resistance to change, which reflected points of implementation vulnerability.

> …at times [it] would demotivate me seeing the lack of engagement from the other staff. (SIL Report, Ward D)

> …when the nursing numbers have been so diminished any extra, anything which seems like it might involve extra work is really resisted (PL, Ward A)

These quotes reflect the relative low priority of PERFECT-ER among staff.

Similarly, organisation-level changes could be slow to materialise leaving ward-level changes difficult to enact in isolation.

> …it takes time for things like documentation groups to approve these things, Trust-wide approaches and they don't want individual approaches on Wards (PL, Ward E)

Lessons learnt from this process evaluation to address consistency in intervention implementation and participant retention for a future definitive trial are reported in figure 3.

## DISCUSSION

PERFECT-ER was developed with the aim of enhancing recovery for people with hip fracture and cognitive impairment by making multiple marginal gains in the quality and consistency of care.[20–22] This process evaluation shows that under certain conditions multiple changes can be implemented leading to multiple marginal gains in quality and consistency and changes in ward ethos. Implementing PERFECT-ER appeared to encourage standardised practices and potentially improved consistency of care for this patient group. Key enablers for implementation were found to be: having dedicated person/s with protected time to drive implementation forward; tapping into existing practices; gaining support from opinion leaders on the ward, using networks and in-ward peer pressure. However, implementation vulnerability occurred when ward pressures were considerable, SILs had limited time and ward staff support was lacking, with marginal gains in care delivery appearing unattainable under these conditions. Overcoming barriers in gaining

dedicated SIL time may be a prime way to enhance future consistency in implementation. Staff availability has been identified as a factor in other process evaluations related to hip fracture patients.[43 45]

Where ward pressures were high and resources for SILs were low, dose suffered. The process evaluation findings suggest that although the intervention was standardised, implementation fluctuated across wards and over time and therefore was not optimal. Consistent with other observations, findings indicate the importance of ward context in implementation.[29–31 46] Ward pressures and inadequate SIL resources contributed to adaptations to the implementation process and sometimes to lack of fidelity in implementation. In response to ward pressures all SILs adapted the implementation process by not conducting ward-level action planning meetings as part of the PDSA cycles. Instead, SILs targeted opinion leaders, using power structures on the wards as mechanisms of impact which may have been stronger to facilitate changes since this approach enabled a successful and streamlined implementation process. Ward staff engagement can be variable even if intervention processes are standardised.[47] This adaptation is likely to have created further losses in ward team engagement, staff reflective practices and commitment and explains why reach was limited in some wards with staff desiring greater involvement with PERFECT-ER implementation. Moving forward, mechanisms for planned staff engagement work should be considered carefully in relation to how and who to engage.

Ongoing interprofessional staff communication in hospitals can take time[48] and relies on individual staff members and supportive organisational structures.[31] In our data, communication work reflected SIL perceptions of self-efficacy through networking with key staff to drive implementation and their employment of peer pressure mechanisms within the ward setting working to obtain compliance. Peer pressure is a component of the CFIR,[42] but in relation to competitiveness with other organisations outside of the inner-ward setting. Our data showed a different type of peer pressure, enacted to maximise compliance within the ward setting using power mechanisms provided by the status of the SIL role itself.

### Strengths and limitations

Our study benefits from multiple sources of data collected from a variety of stakeholders to examine implementation. Data collection involved PPI acting as co-interviewers. Data encompassed several cycles of implementation, obtaining a comprehensive view of the changes over time, processes of implementation, network usage and the role of contextual factors in implementation. Considering the challenges involved when caring for patients with hip fracture and cognitive impairment and the consequent variability in the quality of care,[2 4–7 10–12] our finding that implementing an enhanced recovery checklist appears to encourage standardised practices for this population is promising. Although the findings examined here only relate to five implementation wards, key themes were demonstrated across these and provide a useful starting point for implementation of future complex interventions in acute ward settings and further use of PERFECT-ER.

Limitations to this study include its reliance on SIL-generated data and most patients and carers being unable to compare usual and 'enhanced' care. Patient recruitment for interviews was difficult due to the levels of cognitive impairment, resulting in few patient participants. Most relatives interviewed were one generation younger than the patients, consequently their opinions regarding hospital care may differ. Additionally, staff members who participated in interviews or focus groups may have been those more open to the intervention and to change. The PERFECT-ER intervention required SILs to complete the checklist from individual patient documentary evidence to which researchers did not have access, to provide validation or incorporate assessments of inter-rater reliability. There may have also been inconsistency in intervention scoring. Although participants reported improved standardisation of practices, we cannot ascertain if the indicated implementation reflected change in care delivery, documentation practices or, in some cases, SIL self-implementation. Lastly, this evaluation was of a feasibility, not efficacy trial,[20 23] and it is not yet known whether the changes influence patient outcomes. This may have reduced the motivation of hospital ward staff to implement PERFECT-ER.

### CONCLUSION

Establishing the conditions under which PERFECT-ER was implemented has made visible potential sources of implementation vulnerability such as staffs' perceived priority of the intervention being low, ward pressures and limited improvement lead resources. Our study suggests that due to pressures, most ward staff were not optimally engaged in implementation processes. Rather key senior staff were enlisted as opinion leaders and improvement leads employed peer pressure to further implementation. That key individuals successfully endorsed and enforced implementation indicates that whole-ward interventions may not always require cognitive engagement from all ward staff to implement changes. Future ward-level implementation studies could consider how best to engage staff and which staff groups to target.

**Author affiliations**
[1]School of Health Sciences, University of East Anglia, Norwich, UK
[2]Norwich Medical School, University of East Anglia, Norwich, UK
[3]College of Medicine and Health, University of Exeter, Exeter, UK
[4]School of Education & Lifelong Learning, Univeristy of East Anglia, Norwich, UK
[5]School of Psychology, University of East Anglia, Norwich, UK

**Contributors** JLC, CF, FP, BP, SPH and TB made substantial contributions to the conception or design of the work. SPH, TB and VM-T led on the acquisition of the data. TB, SPH and JLC led the analysis and interpretation of data for the work. TB and JLC led the drafting of the paper. All authors reviewed the paper and gave their final approval of the version to be published. All authors give their agreement

to be accountable for all aspects of the work in ensuring that questions related to the accuracy or integrity of any part of the work are appropriately investigated and resolved. JLC is the guarantor.

**Funding** This work was supported by NIHR Programme Grants for Applied Research (PGfAR) Programme grant number (ref: DTC-RP-PG-0311-12004).

**Competing interests** None declared.

**Patient and public involvement** Patients and/or the public were involved in the design, or conduct, or reporting, or dissemination plans of this research. Refer to the Methods section for further details.

**Patient consent for publication** Not applicable.

**Ethics approval** This study involves human participants and was approved by Camden and Kings Research Ethics Committee (reference number: 16/LO/0621) and Scotland Research Ethics Committee A (reference number: 16/SS/0086). Participants gave informed consent to participate in the study before taking part.

**Provenance and peer review** Not commissioned; externally peer reviewed.

**Data availability statement** Data are available upon reasonable request. Data are available on reasonable request to the corresponding author.

**ORCID iDs**
Tamara Backhouse http://orcid.org/0000-0001-8194-4174
Simon P Hammond http://orcid.org/0000-0002-0473-3610
Jane L Cross http://orcid.org/0000-0002-7003-1916

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
