## [Reviewer comments · BMJ Open]

ARTICLE DETAILS

TITLE (PROVISIONAL)	Implementing an intervention to enhance care delivery and consistency for people with hip fracture and cognitive impairment in acute hospital wards: a mixed methods process evaluation of a randomised controlled feasibility trial (PERFECTED)
AUTHORS	Backhouse, Tamara; Fox, Chris; Hammond, Simon; Poland, Fiona; McDermott-Thompson, Vicki; Penhale, Bridget; Cross, Jane

VERSION 1 – REVIEW

REVIEWER	Nefyn Williams university of liverpool, Health Services Research
REVIEW RETURNED	10-Jul-2022

GENERAL COMMENTS	This is an interesting process evaluation of a feasibility RCT, which follows the MRC framework for process evaluations. The reference for the latest iteration of the MRC framework needs to be updated. The methods state that the process evaluation analysis was informed by CFIR. How was it informed? This needs to be more explicit in the results. I was unclear about the fidelity of the PERFECT-ER intervention. The logic model includes a column about the ward ethos, but I was unclear how much of this had changed. The protocol paper, methods, 1st line of results and discussion all talk about focus groups, but in fact none were performed, why not? I was expecting to see some focus group discussion about any change in the ward ethos. I note that the PERFECT-ER checklist scores did not all increase throughout the successive PDSA cycles. Why was this? Were there particular components that were difficult to implement? Was the intervention aim of making multiple marginal gains in quality and consistency achieved? I was not clear which components of the logic model were realised and which were not. PPI acting as co-interviewers should be included in the strengths part of the discussion. I note that in the main PERFECTED-ER paper it states that mechanisms for delivering consistency in intervention and participant retention need to be addressed. The lessons learnt for a future definitive RCT need to be stated, possibly in a separate box. I could not see in the discussion any comparison with other process evaluations of feasibility trials of hip fracture rehabilitation, such as the FEMuR study.
--

REVIEWER	Brynjar Fure Örebro universitet
REVIEW RETURNED	31-Oct-2022

GENERAL COMMENTS

Thank you for the opportunity to review this manuscript. The topic in question is of interest for multiprofessional teams involved in the care for old people with a hip fracture and cognitive impairment worldwide, both nurses, medical doctors, physiotherapists, occupational therapists, and other professions. In addition, hospital leaders may get knowledge about how to organize care for this patient group.

I have the following inputs to the manuscript:

Title:

The title is long and a bit complicated. Could the title be shortened? Implementing new care for people with hip fracture and cognitive impairment in acute hospital wards

Abstract:

In my opinion, the abstract is representative for the manuscript as a whole. However, I find the first sentence of the conclusion of the abstract diffuse and not very informative. I suggest this sentence is rewritten or removed.

Background:

In the first paragraph of the Background, the authors state that patients with cognitive impairments have complex needs including delirium, behaviors that challenge etc. Do they mean that the patients have complex needs related to delirium, behaviors that challenge etc?

A central part of the PERFECT-ER intervention is, as I understand it, a structured checklist. In the Background, the authors present some key aspects of the intervention, but I cannot find the checklist anywhere in the manuscript. To help readers better understand the PERFECT-ER intervention, I would suggest that the checklist is attached, for instance in supplementary materials.

In the last paragraph of the Background, the aim is presented. Still, I think information regarding how many hospitals that were involved and the method being qualitative and mixed methods belongs in the Methods chapter.

Methods:

The method regarding the qualitative part of the study is, in my opinion, well explained. However, as this study is a mixed methods study, I am a bit surprised that the information regarding analysis of quantitative data based on the checklist is very scarce (only one sentence at the end of the Methods/Analysis chapter). Which quantitative data were analyzed?

Results:

Table 1 presents the persons who were interviewed in this study. My main concern regarding the study is that so few patients (only 3, all female) and partners (only 1 husband) were interviewed. Most of the relatives interviewed are one generation younger than the patients and, therefore, might have opinions regarding hospital care that differ from those of the patients themselves.

In some parts of the results presentation, the results seem to be discussed – for instance under Dose, page 8, line 38.

	In general, I think qualitative results are presented in an informative way. However, I cannot find any reporting of results based on the quantitative data from this study. Is it more correct to refer to this study as qualitative (instead of mixed methods)? Discussion: As I understand it, it is not yet known whether the PERFECT-ER intervention is effective or not. My main input to the discussion is: Is it possible to motivate nurses, doctors and therapists in hospital wards to implement an intervention without knowing whether the intervention is effective and leads to improved clinical outcomes for the patients? It could be argued that it would be more useful to perform this study at a later stage when the potential effectiveness of the intervention has been clarified. In my opinion, this aspect should be emphasized more clearly in the discussion. Conclusion: Again, I suggest replacing the phrase “existing and new power structures within the ward were used to facilitate implementation” by more useful, concrete and precise information on the exact power structures that were used
--	---

VERSION 1 – AUTHOR RESPONSE

Reviewer: 1

Prof. Nefyn Williams, university of liverpool Comments to the Author:

This is an interesting process evaluation of a feasibility RCT, which follows the MRC framework for process evaluations.

The reference for the latest iteration of the MRC framework needs to be updated.

RESPONSE: Many thanks, this reference has been added.

The methods state that the process evaluation analysis was informed by CFIR. How was it informed? This needs to be more explicit in the results.

RESPONSE: The CFIR informed our analysis and is embedded implicitly throughout the results section related to the inner setting (hospital ward), characteristics of individuals (particularly SILs) and process. For example, we examine implementation in relation to these specific aspects of the CFIR: **INNER SETTING DOMAIN:** networks and communication, implementation climate (relative priority of PERFECT-ER), readiness for implementation (leadership engagement, available resources, and access to information).

CHARACTERISTICS OF INDIVIDUALS DOMAIN: self-efficacy of SILs

PROCESS DOMAIN: planning, engagement (opinion leaders - in regard to Ward Managers and Process leads and senior ward staff, formally appointed internal implementation leaders represented by SILs, and champions), executing (e.g., fidelity and mechanisms of impact).

We do take on board your point that these are not explicit in the results section since we structured the results in line with the MRC guidance categories. Therefore, for this revision, we have tried to incorporate the same terminology as the CFIR throughout the results as much as possible to make clear relevant aspects from the framework. We have also specified in the methods section, the relevant areas of the CFIR for our analysis focus:

‘The Consolidated Framework for Implementation Research (CFIR)[43] in regard to aspects of the inner setting (hospital ward), characteristics of individuals, and process.’

I was unclear about the fidelity of the PERFECT-ER intervention. The logic model includes a column about the ward ethos, but I was unclear how much of this had changed.

RESPONSE: Many thanks, for highlighting this. Ward ethos had changed in those wards where implementation went well. More information has been added to the fidelity results section to address changes, or not, in ward ethos:

'Changes in ward ethos varied and appeared to be linked to the implementation climate, dose delivered and reach. Staff at ward C, where implementation went well, talked about changes in approach.

Occupational Therapist: *I think as [senior nurse] highlighted, it's [PERFECT-ER] brought an awareness of delirium... I have had a sense of that amongst the nursing colleagues ...people are a lot more sensitive and patient and maybe provide better care maybe.*

Senior Nurse: *Yeah I agree*
(Focus Group, Ward C)

Some participants indicated PERFECT-ER provided a guiding philosophy for staff.

PERFECT-ER is our ward philosophy do you know what I mean? ... it actually is part of their [Ward staff's] umm vision part of their thinking about their patient care (SIL, Ward A)

Whereas, in Ward E which struggled to implement, there was no ethos change.

Occupational Therapist: *As far as the PERFECTED Ward goes I am not sure how much it has really, it has sort of changed our practice as such because we kind of work to enable people anyway...*

Lead Occupational Therapist: *Yeah, I am still not a hundred percent clear about the different check lists ...that makes it the PERFECTED Ward ...I don't think it is something that is obvious what we are working towards.*
(Focus Group, Ward E)

And in the discussion:

'This process evaluation shows that under certain conditions multiple changes can be implemented leading to multiple marginal gains in quality and consistency and changes in ward ethos... Key enablers for implementation were found to be: having dedicated person/s with protected time to drive implementation forward; tapping into existing practices; gaining support from opinion leaders on the ward, utilising networks and in-ward peer pressure. However, implementation vulnerability occurred when ward pressures were considerable, SILs had limited time and ward staff support was lacking, with marginal gains in care delivery appearing unattainable under these conditions.'

The protocol paper, methods, 1st line of results and discussion all talk about focus groups, but in fact none were performed, why not? I was expecting to see some focus group discussion about any change in the ward ethos.

RESPONSE: As shown in Table 1, 24 hospital staff and 2 research nurses were involved in focus groups. We have edited the quotations codes to make it clear when data are derived from focus groups. For example, (Focus Group, Ward E).

I note that the PERFECT-ER checklist scores did not all increase throughout the successive PDSA cycles. Why was this?

RESPONSE: This is now covered more clearly in the Dose section:

'Availability of resources appeared to help or hinder implementation. Wards B and C implemented well (PERFECT-ER scores increasing 22% and 12% respectively over the study); these sites had dedicated SIL time and clear ward processes which could be utilised for implementation. Wards A (PERFECT-ER scores increasing 8%), D (PERFECT-ER scores decreasing 7%) and E (PERFECT-ER scores decreasing 9%) struggled with severe ward pressures, including staff shortages, affecting implementation. Wards D and E also had difficulties providing dedicated SIL time and the implementation climate reflected a lack of consistent support from ward staff.'

Were there particular components that were difficult to implement?

RESPONSE: This information has been added to the Dose section:

'PERFECT-ER practice items least implemented were: 1) carers given the opportunity to accompany the patient in the recovery room, and 2) pain assessed daily using a pain scale specific to patients with cognitive impairment'

Was the intervention aim of making multiple marginal gains in quality and consistency achieved? I was not clear which components of the logic model were realised and which were not.

Many thanks for highlighting this. We have now made this clear in the discussion section:

'This process evaluation shows that under certain conditions multiple changes can be implemented leading to multiple marginal gains in quality and consistency and changes in ward ethos. Implementing PERFECT-ER appeared to encourage standardised practices and potentially improved

consistency of care for this patient group. Key enablers for implementation were found to be: having dedicated person/s with protected time to drive implementation forward; tapping into existing practices; gaining support from opinion leaders on the ward, utilising networks and in-ward peer pressure. However, implementation vulnerability occurred when ward pressures were considerable, SILs had limited time and ward staff support was lacking, with marginal gains in care delivery appearing unattainable under these conditions.'

PPI acting as co-interviewers should be included in the strengths part of the discussion.

RESPONSE: Many thanks, this has been added to the strengths section.

I note that in the main PERFECTED-ER paper it states that mechanisms for delivering consistency in intervention and participant retention need to be addressed. The lessons learnt for a future definitive RCT need to be stated, possibly in a separate box.

RESPONSE: A new box has been created, outlining the lessons learned for a definitive trial. Please see Figure 3 below:

PERFECTED Process Evaluation: Lessons Learned

Delivering consistency in intervention

- Protected Service Improvement Lead time must be ensured, consider staff resource when recruiting sites
- If there is more than one Service Improvement Lead at a hospital site, each must have their own responsibilities/accountability
- Service Improvement Leads should be hired from the study ward or more time and support given to those external to the ward
- Consider administrative support for Service Improvement Leads (ward clerk time)
- Create a generic PERFECT-ER leaflet to provide a basic level of information about the intervention for ward staff and provide Service Improvement Lead's name and contact
- Provide Process Leads with role definition, clear responsibilities and make time allocation transparent
- Refine PERFECT-ER Manual to reflect feedback and role of opinion leaders in implementation

Participant recruitment and retention

- Research team to use a problem solving approach with Process Leads and research nurses to support recruitment and retention of participants
- Provide dementia training for research nurses
- Colour-code participant documents to reduce confusion
- Enable relative participants to fill in Case Report Forms/workbooks in own time rather than at the hospital – possible online option
- Reduce number of outcome measures to decrease burden on relatives and patients
- Consider care-home staff as informants if patients are discharged to a care home and relatives can no longer answer questions about the person's daily life

Figure 3: PERFECTED: Lessons Learned for a Future Definitive Randomised Controlled Trial

I could not see in the discussion any comparison with other process evaluations of feasibility trials of hip fracture rehabilitation, such as the FEMuR study.

RESPONSE: We had referred to other process evaluations involving hospital ward staff practice (Bridges et al., [31]; Sheard et al., [46]; Kummerow et al., [47]). However, many thanks for highlighting this, we have now added the FEMuR study process evaluation (Roberts et al., [44]).

Reviewer: 2

Dr. Brynjar Fure, Örebro universitet

Comments to the Author:

Thank you for the opportunity to review this manuscript. The topic in question is of interest for multiprofessional teams involved in the care for old people with a hip fracture and cognitive impairment worldwide, both nurses, medical doctors, physiotherapists, occupational therapists, and other professions. In addition, hospital leaders may get knowledge about how to organize care for this patient group.

I have the following inputs to the manuscript:

Title:

The title is long and a bit complicated. Could the title be shortened?

Implementing new care for people with hip fracture and cognitive impairment in acute hospital wards

.....

****Comment from the editor: While we understand the reviewers comment our inhouse style is that titles must include include the research question, study design and setting.****

RESPONSE: Many thanks, although we agree the title is long, we have left it the same to comply with journal style rules.

Abstract:

In my opinion, the abstract is representative for the manuscript as a whole. However, I find the first sentence of the conclusion of the abstract diffuse and not very informative. I suggest this sentence is rewritten or removed.

RESPONSE: Many thanks for highlighting this. The sentence now reads: *'Our study suggests that implementation was expediated when senior staff were onboard as opinion leaders and formally appointed internal implementation leaders exerted their power.'*

Background:

In the first paragraph of the Background, the authors state that patients with cognitive impairments have complex needs including delirium, behaviors that challenge etc. Do they mean that the patients have complex needs related to delirium, behaviors that challenge etc?

RESPONSE: many thanks, *'including'* has been changed to *'related to'*.

A central part of the PERFECT-ER intervention is, as I understand it, a structured checklist. In the Background, the authors present some key aspects of the intervention, but I cannot find the checklist anywhere in the manuscript. To help readers better understand the PERFECT-ER intervention, I would suggest that the checklist is attached, for instance in supplementary materials.

RESPONSE: Many thanks for this comment. The PERFECT-ER Intervention (checklist) will soon be in the public domain when the NIHR Programme Grant for Applied Research final report is published (forthcoming early in 2023). We have added a reference to this forthcoming publication into this manuscript.

In the last paragraph of the Background, the aim is presented. Still, I think information regarding how many hospitals that were involved and the method being qualitative and mixed methods belongs in the Methods chapter.

RESPONSE: Many thanks, this information has been moved from the background to the first part of the methods section.

Methods:

The method regarding the qualitative part of the study is, in my opinion, well explained. However, as this study is a mixed methods study, I am a bit surprised that the information regarding analysis of quantitative data based on the checklist is very scarce (only one sentence at the end of the Methods/Analysis chapter). Which quantitative data were analyzed?

RESPONSE: Many thanks for your comment. We describe the quantitative data earlier in the methods section:

'PERFECT-ER checklist scores were obtained from patient notes by the SILs. At each ward there were eight checklist time-points: 3 months pre-trial, 1.5 months pre-trial, trial baseline, 4 months, 7 months, 10 months, 13 months, and end of trial (15 months).'

We have added to the sentence in the analysis section, which now reads:

'Quantitative data from PERFECT-ER checklist scores, collected from five patients at each of the five sites at eight timepoints, were analysed descriptively by ward and implementation cycle.'

Figure 1 describes the quantitative data and Figure 2 provides the results of the quantitative data analysis. We have also added percentage increases or decreases to the dose section:

'Wards B and C implemented well (PERFECT-ER scores increasing 22% and 12% respectively over the study); these sites had dedicated SIL time and clear ward processes which could be utilised for implementation. Wards A (PERFECT-ER scores increasing 8%), D (PERFECT-ER scores decreasing 7%) and E (PERFECT-ER scores decreasing 9%) struggled with severe ward pressures, including staff shortages, affecting implementation. Wards D and E also had difficulties providing dedicated SIL time and the implementation climate reflected a lack of consistent support from ward staff.'

Results:

Table 1 presents the persons who were interviewed in this study. My main concern regarding the study is that so few patients (only 3, all female) and partners (only 1 husband) were interviewed. Most of the relatives interviewed are one generation younger than the patients and, therefore, might have opinions regarding hospital care that differ from those of the patients themselves.

RESPONSE: Many thanks for highlighting this. Recruitment of patients was difficult due to their cognitive impairment and often patients were widowed so supported by those from the younger generation. We have added these factors to our limitations section:

'Patient recruitment for interviews was difficult due to levels of cognitive impairment, resulting in few patient participants. Most relatives interviewed were one generation younger than the patients, consequently their opinions regarding hospital care may differ.'

In some parts of the results presentation, the results seem to be discussed – for instance under Dose, page 8, line 38.

RESPONSE: The discussion aspect on line 38 (Dose section) has been deleted.

In general, I think qualitative results are presented in an informative way.

However, I cannot find any reporting of results based on the quantitative data from this study. Is it more correct to refer to this study as qualitative (instead of mixed methods)?

RESPONSE: The study was mixed methods, albeit predominantly qualitative. The results from the quantitative data are presented in Figure 2 and reported in the Dose section (excerpt in response above).

Discussion:

As I understand it, it is not yet known whether the PERFECT-ER intervention is effective or not.

My main input to the discussion is: Is it possible to motivate nurses, doctors and therapists in hospital wards to implement an intervention without knowing whether the intervention is effective and leads to improved clinical outcomes for the patients? It could be argued that it would be more useful to perform this study at a later stage when the potential effectiveness of the intervention has been clarified. In my opinion, this aspect should be emphasized more clearly in the discussion.

RESPONSE: Many thanks, we have added this aspect to our limitations section:

'Lastly, this evaluation was of a feasibility, not efficacy trial[20,23], and it is not yet known whether the changes influence patient outcomes. This may have reduced the motivation of hospital ward staff to implement PERFECT-ER.'

Conclusion:

Again, I suggest replacing the phrase “existing and new power structures within the ward were used to facilitate implementation” by more useful, concrete and precise information on the exact power structures that were used

RESPONSE: Many thanks for your suggestion. We have reviewed and deleted this sentence since the sentence before outlined the precise information (opinion leaders and peer pressure).

VERSION 2 – REVIEW

REVIEWER	Nefyn Williams university of liverpool, Health Services Research
REVIEW RETURNED	21-Dec-2022

GENERAL COMMENTS	I am happy with the changes that you have made to the paper.
--

REVIEWER	Brynjar Fure Örebro universitet
REVIEW RETURNED	07-Jan-2023

GENERAL COMMENTS	Thank you for the revised version of the manuscript. In my opinion, the manuscript has improved after the revision based on inputs from the reviewers, in particular regarding limitations of the study and more clear conclusions. The manuscript is of interest for health care leaders, nurses, doctors and therapists working with patients with hip fractures within different specialties, notably within geriatric and orthopedic medicine. In my view, this manuscript can be published in its present version.
---